# Facultative Annual Life Cycles in Seagrasses

**DOI:** 10.3390/plants12102002

**Published:** 2023-05-16

**Authors:** Marieke M. van Katwijk, Brigitta I. van Tussenbroek

**Affiliations:** 1Department of Environmental Science, Radboud Institute of Biological and Environmental Sciences, Radboud University, Heyendaalseweg 135, 6525AJ Nijmegen, The Netherlands; 2Institute of Ocean Sciences and Limnology, Universidad Nacional Autónoma de México, Puerto Morelos 77580, Mexico; vantuss@cmarl.unam.mx

**Keywords:** life history, sexual reproduction, *Halophila*, *Ruppia*, *Zostera*, *Oryza*, perennial

## Abstract

Plant species usually have either annual or perennial life cycles, but facultative annual species have annual or perennial populations depending on their environment. In terrestrial angiosperms, facultative annual species are rare, with wild rice being one of the few examples. Our review shows that in marine angiosperms (seagrasses) facultative annual species are more common: six (of 63) seagrass species are facultative annual. It concerns *Zostera marina*, *Z. japonica*, *Halophila decipiens*, *H. beccarii*, *Ruppia maritima*, and *R. spiralis*. The annual populations generally produce five times more seeds than their conspecific perennial populations. Facultative annual seagrass species occur worldwide. Populations of seagrasses are commonly perennial, but the facultative annual species had annual populations when exposed to desiccation, anoxia-related factors, shading, or heat stress. A system-wide ‘experiment’ (closure of two out of three connected estuaries for large-scale coastal protection works) showed that the initial annual *Z. marina* population could shift to a perennial life cycle within 5 years, depending on environmental circumstances. We discuss potential mechanisms and implications for plant culture. Further exploration of flexible life histories in plant species, and seagrasses in particular, may aid in answering questions about trade-offs between vegetative and sexual reproduction, and preprogrammed senescence.

## 1. Introduction

Facultative annual species are perennial species that have some populations displaying annual life histories under certain conditions, completing their life cycles from germination to seed production followed by death within one year. Although it is known that the lengths of the life cycles of plants can vary depending on the environment, especially in biennials [1], facultative annual life histories are uncommon for angiosperms. To our knowledge, it is described in only two, unrelated, terrestrial species: wild rice *Oryza perennis* [2] and the herb *Erythrante guttata* (syn. *Mimulus guttatus* [3]). Facultative perennials, i.e., annual plants that may have perennial populations, are also uncommon [4,5]. The marine environment is colonized by a few angiosperm species that are all clonal and perennial [6] (Appendix A), but annual populations of the well-investigated seagrass species *Zostera marina* were already described in the 1970s [7]. From a biological and ecological perspective, such a flexible life-cycle strategy is an interesting phenomenon worth exploring.

The marine environment has posed special challenges to angiosperms which required physiological and reproductive adaptations [8,9]. Possibly, flexible life cycles are another, until now underexplored, response to the marine environment, which we wish to address here. We study whether other (perennial) seagrass species, in addition to the well-studied *Z. marina*, present annual populations. Secondly, we review the varying environmental settings of facultative annual seagrasses. In this review, we specifically question the following. (1) How common is facultative annual life history among seagrass species? (2) Is the occurrence of facultative annual life history widespread geographically? (3) Does the annual seed production differ with life cycle lengths? (4) Is there a relationship between life history and environmental settings? (5) Is there evidence for shifts between life histories?

## 2. A Facultative Annual Life History Is Widespread among Seagrass Species and Occurs Worldwide

Literature review shows that there are no true annual seagrass species. An annual life cycle was suggested by Kuo et al. [10] for the understudied deep water dioecious *Halophila tricostata*, but recent work by Chartrand [11] showed that this species overwintered with quiescent rhizomes, although yearly recurring seedling recruitment was important for persistence. Similar life history strategies with vegetative quiescent phases have been revealed for other seagrass species (Appendix A).

Based on available evidence, at least 6 out of 63 seagrass species display a facultative annual life history, with true annual populations, namely *Zostera marina*, *Z. japonica*, *Halophila decipiens*, *H. beccarii*, *Ruppia maritima*, and *R. spiralis* (Appendix A). The trait is polyphyletic, as these species belong to different families (Hydrocharitaceae, Ruppiaceae, and Zosteraceae) [12]. *Z. marina* is the best-known facultative annual seagrass species. This species occurs in the temperate and tropical northern hemispheres, with annual populations recorded at several locations (Figure 1).

## 3. Seed Production Is Higher in Annual Than Perennial Populations

The seed production of annual populations tends to be higher than that of perennial populations in the six facultative annual seagrass species identified in our study (Figure 2). Overall, seed production is five times higher in annual populations compared to conspecific perennial populations. However, populations vary greatly in seed production, and some perennial populations also present high seed outputs; for example, a perennial population of *Z. marina* in Chesapeake Bay had a potential maximum seed production of 40,000 seeds/m^2^ [13] vs. 100,000 seeds/m^2^ of an annual population in the subtropical Gulf of California [14]. Annual *Z. marina* plants typically have limited rhizome development and allocate most of the aboveground biomass to reproductive shoots [7]. Such differential allocation to vegetative and reproductive structures has been found for terrestrial angiosperms when comparing annual and perennial congeneric species [4,15,16].

## 4. Annual Populations Live in More Stressful Environments

Assuming that there is a trade-off between vegetative (clonal) growth and sexual reproduction [16,17,18] and that sexual reproduction competes with the vegetative functions for necessary resources for plant growth and maintenance, an annual life cycle should only be favored over a perennial cycle when the survivorship of the established plant is lower than that of the seed or seedling. Such unfavorable conditions for vegetative development may recur periodically (often seasonally) or at stochastic intervals in highly unpredictable environments [19].

In seagrasses, such periodically unfavorable conditions may be low temperatures combined with high turbidity, as was found in *Zostera japonica* (British Columbia [20]) and *Ruppia maritima* (Baltic Sea [21]). *Ruppia* spp. may colonize shallow coastal lagoons that are only flooded during part of the year, and annual growth forms are reported to be a response to desiccation (Appendix A). *Halophila beccarii* forms annual populations as a response to decreased salinities on tidal flats in Malaysia [22]. Additionally, the subtidal delicate and shallow-rooted *Halophila decipiens* does not have a broad tolerance to salinity or temperature changes and may therefore be susceptible to removal or die-off during winter (Appendix A).

Annual populations of the relatively well-studied *Z. marina* are encountered in a myriad of situations (Figure 3 and Box 1). Comparing habitats of annual populations with the nearest perennial ones, the first seems to be more stressful than the latter. They experience either desiccation, heat stress, anoxia-related stress, shading stress, or a combination of all these well-known stressors of *Z. marina* and other seagrass species [23]. Populations are usually annual in the intertidal, where they experience periodic desiccation, but in water-retaining depressions and in moist air intertidal, plants have a perennial live history (Box 1). Subtidal or submersed annual populations seem to be exposed to higher levels of anoxia compared to those in neighboring populations (Box 1). Anoxia-related stress includes excessive eutrophication and/or organic matter loading, at times accompanied by lower salinity (as a covariate of enhanced nutrient input from freshwater sources), increased shading, warmer circumstances (decreasing dissolved oxygen and likely enhancing microbial processes leading to anoxia), or muddier sediments (mud is often correlated with organic matter and occurs in areas with less flushing). Anoxia results in the microbial production of sulfide and ammonia, which are toxic to *Zostera* spp. [24,25]. In addition, tidal or submersed annual populations occur in heat-stressed environments and in light-limited (deep) habitats (Box 1).

Box 1In what environments can we find annual and perennial populations of *Zostera marina*?Annual populations of *Zostera marina* may recur at the same sites for decades, without perennial neighbors [27] (Appendix A), and thus are likely self-sustaining.Perennial populations can be encountered as follows:In subtidal or submersed environments with low or moderate eutrophication (this is the typical environment and life cycle for *Z. marina*);Exceptionally, in mid-intertidal environments that
probably remain sufficiently moist during low tide, namely (a) in tidal pools
where the plants remain submersed (US [28,29,30]; probably NW Europe [31]) and (b) where high air moisture during the growing
season (humid climate and sea mists) may protect the plants from desiccation
in the mid-intertidal zone: along the eastern shores of the UK and Ireland [32,33]; *Z. marina*
is here referred to as *Z. angustifolia*), and probably also along the
Southwest coast of US, as suggested by the low flowering frequency (33% in
Carlsbad [28]), and the robust perennial growth form encountered in San Diego, pers. obs. first author);Even more exceptionally, in coarse sanded mid-intertidal
areas, at a slightly higher tidal level than the nearby annual population,
where they experience even more desiccation. They lose aboveground biomass
during summer as a consequence, but rhizomes survive both during summer and
winter, the latter likely due to the coarse sediments that allow for flushing
(observed in the southern and northern Wadden Sea [34]).Annual populations can be encountered as follows:4.In mid-intertidal environments that are twice-daily
exposed to air on the east and west coast of North America and in NW Europe.
All seedlings may develop into reproductive shoots [7], or, alternatively,
a consistent part of the population may consist of vegetative shoots during
the growing season, but they disappear (including belowground parts) during
winter (e.g., in Zandkreek, Europe [35,36]). In North America (both east and west coast),
transitions from annual to perennial populations coincide with the tidal
depth gradient; from the mid-intertidal towards the low tide level, an
increasing number of plants becomes perennial [7,29,30];5.Permanently submersed environments on the east coast of
the USA, in NW Europe, Japan, and Korea, with muddier, more turbid, warmer,
more eutrophicated, and/or less saline conditions as compared to those of
nearby perennial populations [37,38,39]. Generally, not all shoots are reproductive; some shoots
are vegetative and may last longer than the reproductive shoots until they
finally disappear (including belowground parts) during winter [13,36]. These
populations may represent a transition between perennial and annual life
histories;6.Deep submersed environments where light is limiting.
Nearby perennial populations are located shallower, described for Korea [40] and NW Europe [41];7.Permanently submersed environments with yearly recurrent
heat stress. There are no perennial populations nearby, described for several
populations in the Gulf of California, at the southern distribution limit of
this species. All shoots of these plants become reproductive [42].
Note: Some populations are called ‘annual’ or a separate
ecotype but seem to occupy marginal habitats incidentally colonized by
incoming seed from nearby populations; thus, they are not self-sustaining
populations [43,44].

## 5. Shifts between Annual and Perennial Life Histories in *Zostera marina*

System scale ‘experiments’ in the Southwest Netherlands have shown that annual populations can become perennial within 5 years after a change in environment. Three estuary branches were modified for coastal protection during 1961–1986: one branch was modified into an oligotrophic saline lake [45], one branch was modified into a brackish and eutrophic lake [46], whereas one branch remained intertidal with a modified hydrodynamic regime [47]. Prior to the modifications, the branches were connected, and they all hosted intertidal, annual populations of *Z. marina* [48]. In the newly formed oligotrophic saline lake, the population became perennial upon submergence within 5 years [41]. However, in the newly formed brackish and eutrophic lake as well as the intertidal branch, the populations continued to be annual ([36,39]; Figure 4). This shift in life history, or absence thereof, after modification of the environment, is evidence that population life history traits can be induced by the environment. When the plants became perennial, they presented lower seed production and a number of flowering shoots, higher belowground biomass, and the vegetative shoots showed vigorous growth earlier in the season than before, when the population was still annual and seasonal timing is earlier, suggesting that rhizomes give the shoots a head start as compared to the seed [36].

Transplantation experiments in NW Europe and in North America confirm that seedlings from annual populations can become perennial plants during the first winter (NW Europe [34], Izembek Lagoon, Alaska [49], although their reproductive effort remains high (NW Europe [34], Willapa bay, Washington [50]). Keddy and Patriquin [7] cultivated seedlings in the laboratory from seeds originating from annual and perennial populations in Nova Scotia and found that 28 out of 29 of the seedlings from the ‘annual’ seeds developed into annual plants and 1 developed into a perennial plant. Vice versa, 26 of 28 seedlings from ‘perennial’ seeds developed into perennial plants, whereas 2 of 28 developed into an annual plant. Thus, the findings of Keddy and Patriquin [7] suggest that annual populations have the potential to produce perennial offspring and vice versa.

It is intriguing that the seedlings of the reviewed annual populations produce reproductive shoots very early in development; in other words, they are “programmed for scenescence” several months later. Secondly, it is intriguing that they, nevertheless, may shift to a perennial life history when the environment becomes more favorable for vegetative survival in critical periods.

## 6. What Mechanisms May Induce an Annual or Perennial Life Cycle? Future Avenues of Research

During early growth, the seedlings of annual *Zostera marina* plants may not receive any indications from their environment that they will encounter adverse conditions for perennial growth later in the season, and the rapid development of generative shoots and early scenescence are perhaps “programmed”. Chartrand [11] found indications for such programming in deep water annual populations of *Halophila decipiens* in tropical Australia. However, it is also possible that a more stressful environment may already manifest early in the season and induce lower productivity/respiration ratios in the seedlings (see Figure 3 and Box 1). This lower P/R ratio may induce the plant to invest more resources into sexual reproduction, which is also suggested by a review of the effect of disturbance on sexual reproduction in seagrasses by Cabaço and Santos [51], and supported by later studies, for example, showing relations between sexual reproductive effort and temperature [52,53,54], but see [55], desiccation [43,56], nutrients [57,58], mechanical disturbance [59], and high salinity [60].

Population genetic studies in NW Europe [61] and in San Francisco Bay US [62] suggest a lack of genetic differentiation between annual and perennial populations, as well as high rates of gene flow between them, although genetic diversity is generally larger in the annual than in perennial populations [63]. Muñoz-Salazar and coworkers [64] found significant genetic differentiation between perennial *Z. marina* populations from the Pacific coast and annual ones in the Gulf of California (the summer annuals, type number 1 in Figure 1). This genetic divergence may be explained by the different life histories (annual vs. perennial), but it could also have been generated by limited gene flow between the two regions, as the tropical waters and current patterns of the southern Gulf of California have presented a barrier to gene flow and migration since the end of the Pleistocene. Oetjen and coworkers [65], using a genome scanning approach (using SNP and microsatellite markers), found some indications of selection between the subtidal perennial and intertidal annual populations in NW Europe. Divergent selections between the types of populations were detected at six loci, of which three were linked to genes involved in osmoregulation, water balance, and sexual reproduction (seed maturation). Selection could be enhanced by the different timing of the flowering initiation, even if annual populations are located in the immediate proximity of perennial populations via reproductive isolation [18,66].

Our review suggests that the annual vs. perennial life cycles in facultative annual *Z. marina* (and possibly the other facultative annual seagrass species) may be reversible, involving tradeoffs between vegetative and generative functions. Genetic evidence of such inflection of tradeoff was, for example, found in the terrestrial annual *Arabidopsis thaliana*. Modulation of the activities of only three genes influenced the indeterminacy of meristems and longevity of the plants, resulting in a growth form with the increasing development of vegetative buds, higher longevity, and extensive woodiness, indicative of perennial plants [67]. In the two terrestrial facultative annuals described in the literature, *Erythrante guttata* and *Oryza sativa*, possible genetic mechanisms for such reversibility between life histories have been investigated. Friedman and coworkers [3], when identifying phenotypic and genetic trade-offs between flowering and vegetative growth in *E. guttata*, found that differential responses to photoperiod and vernalization (the induction of a plant’s flowering process by exposure to the prolonged cold of winter) of plants from annual or perennial populations involved quantitative trait loci (QTL) and differential gene expression. QTL was also found to influence resource allocation in annual and perennial populations of rice *Oryza* [68,69].

In general, gene expression may be involved in frequent and precocious flowering. Perennial plants require reprogramming of some meristems to start the production of reproductive organs. Overexpression of the Flowering Locus (FT) gene from *A. thaliana* resulted in precocious flower development independent of photoperiod [70]. In the same plant, it was found that micro RNAs are involved in gene expression; miRNA 172 (miR172) caused early flowering through disruption of the downregulation of floral repression genes [71]. Interspecies gene transfer between perennial *Arabis alpina* and *A. thaliana*, showed a perennial and an annual signaling pathway to flowering, involving Squamosa promotor binding protein-like 15 (SPL15) and FL pathways, respectively [72]. The functional overlap between the pathways may enable flexible responses to shifting environments, as well as life history variation [72]. In general, from an evolutionary perspective, life history traits are among the most labile trait syndromes in flowering plants and annuality has evolved convergently in different lineages of flowering plants, though mechanisms underlying transitions are still unclear [16].

Chartrand [11] found that the general condition of the seagrass plants of the deep-water annual population of the seagrass *H. decipiens* declined before the light levels fell below the critical threshold for growth, from which she suggests that senescence and sexual reproduction were programmed. She observed shifts in hormones involved in these processes similar to shifts previously reported in terrestrial plants [4]. Up- and downregulation of corresponding areas could be confirmed with metabolomic profile analysis. Such changes in metabolomic expression may be heritable (epigenetic); epigenetic changes may last through cell divisions for the duration of the plant’s life and may also last for multiple generations, even though they do not involve changes in the underlying DNA sequence of the organism [73]. In short, annual life cycles in facultative annual species seem to be induced by the environment (for example, by low P/R ratios) or (epi-) genetic programming. Further research into mechanisms that induce the annual life cycle is needed, and the six seagrass species detected in this review may be good candidates for such studies.

## 7. Perspectives for Plant Culture

The tradeoff between investment in generative or vegetative plant parts becomes visible and tangible in facultative annual species, making this type of species of interest for plant biological and ecological studies. Moreover, when seagrass plants are to be cultured at a large scale, for example, to rewild the sea with domesticated seagrass when donor populations are scarce [74], knowing the factors that determine the life history may help to maximize seed production. Though seagrass domestication is still in its infancy, the facultative annual live strategy may allow a culture aiming at a balance between seed production and whole-year maintenance of ecosystem services provided by vegetative plant parts typically provided by perennial populations (such as carbon sequestration and erosion control). Such an ‘ideal’ tradeoff is presently targeted in terrestrial crops such as wheat and rice for a more sustainable agricultural practice, where perennial strains or species are domesticated to not only maintain food security but also ecosystem services such as erosion control and improved nitrogen use efficiency [4,75,76,77,78,79,80]. It is probably not a coincidence that perennial rice development, being one of the rare terrestrial facultative annual species, is more advanced than the development of perennial wheat, which requires de novo domestication of a congeneric species [78]. Our review shows that facultative annual species could potentially be brought to an optimum of seed production and vegetative development. Further research should elucidate whether this could be accomplished via manipulation of stresses or through other means such as (epi)genetic selection. Considering this, the study of facultative annual seagrasses may reveal a “rice from the sea” in the future.

## Figures and Tables

**Figure 1 plants-12-02002-f001:**
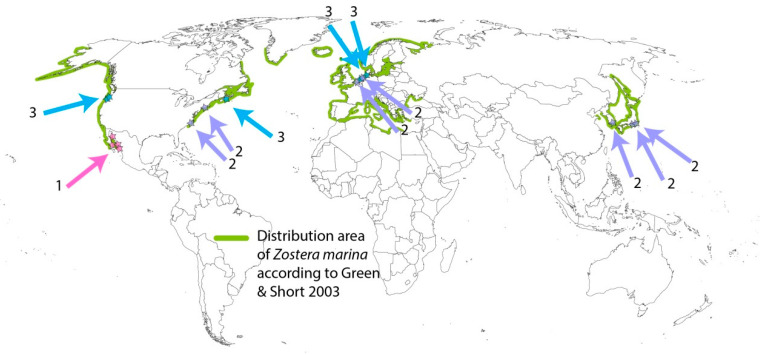
Map of locations of annual *Zostera marina* populations (arrows) and the total distribution of *Z. marina* (green coastlines). Numbers indicate three types of environments. 1: Environments with yearly recurrent heat stress. 2: Subtidal or permanently submersed environments experiencing anoxia-related stress or shading stress. 3: Mid-intertidal environments with twice-daily exposure to air. More explanation in Section 4 and Appendix A.

**Figure 2 plants-12-02002-f002:**
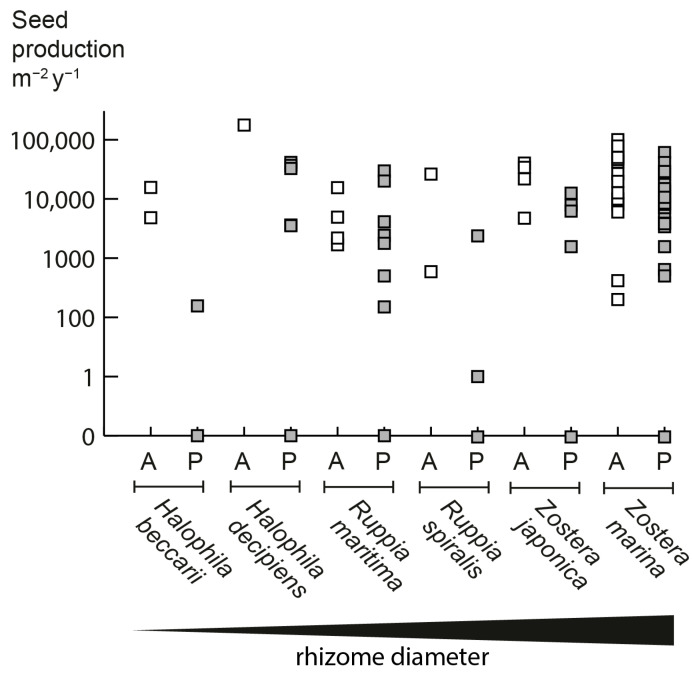
Annual seed production in the six facultative annual seagrass species. A = Annual (open symbols), P = Perennial (grey symbols). If a range was presented, both minimal and maximal values are indicated. Please note that the perennial populations of all species can also have zero seed production, which is quite common for seagrasses. Literature sources and data are in Appendix A.

**Figure 3 plants-12-02002-f003:**
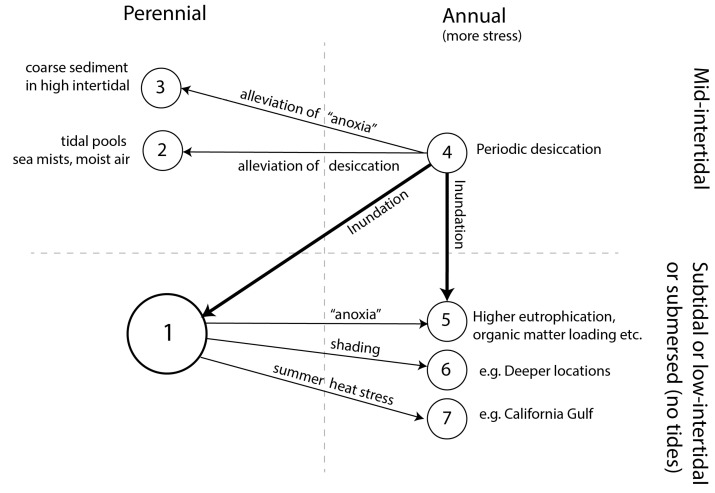
Overview of reported habitat characteristics of annual and perennial *Zostera marina* populations and alleged drivers for transitions between life histories (arrows). The numbers refer to types of reported habitats listed in Box 1. Number 1 is the most typical growth strategy and environment for *Z. marina*. Thin arrows depict relative differences between environments of annual populations versus nearest perennial populations (correlative). Thick arrows refer to a system-scale inundation “experiment”; see Section 5. The term “anoxia” refers to a situation of more nutrient loading and/or muddier or more organic sediments and/or more anoxic sediment, often accompanied by higher turbidity and lower salinity due to freshwater origin of the nutrient or organic matter loads. These factors usually covariate in eutrophic situations [26].

**Figure 4 plants-12-02002-f004:**
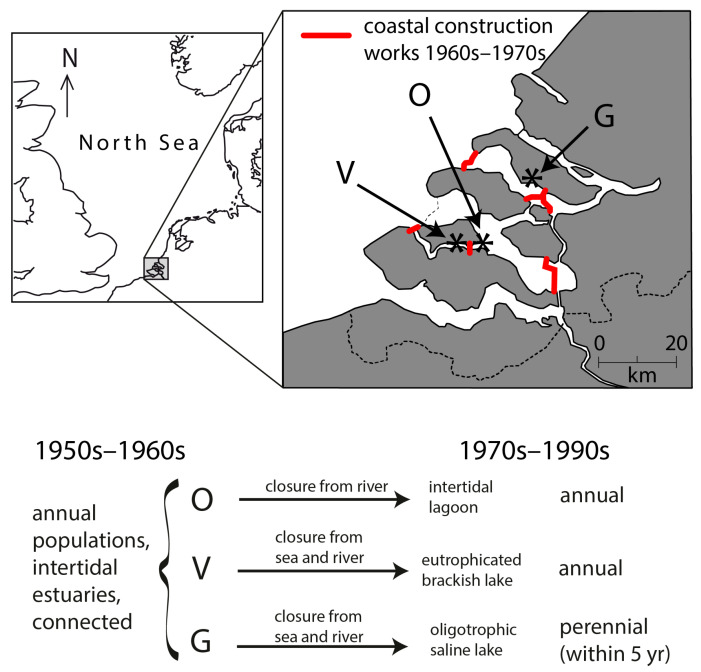
Dutch waterworks resulting in an unintended system-scale “experiment” in the southwest Netherlands. Prior to coastal defense works, three arms of the Meuse-Rhine estuary harbored annual, mid-intertidal *Zostera marina* [48]. During the 1960s and 1970s, two of the branches were closed, forming a brackish lake (V = Veere) and a saline lake (G = Grevelingen), whereas the third arm (O = Oosterschelde) remained tidal, though cut off from river water. The freshwater input varied between the lakes; as a result, the brackish Lake Veere had a low salinity, high nutrient loading, macroalgal blooms, high turbidity, and periods of anoxia, whereas the saline Lake Grevelingen had a higher salinity, lower nutrient loading, higher water clarity, and lower algal growth [39]. In the Oosterschelde and the eutrophic brackish lake Veere, *Z. marina* plants remained annual, but in the oligotrophic saline lake Grevelingen, the plants became perennial within 5 years (comparing [48] with [36,41]). All populations went (near) extinct during the last 3 decades.

## Data Availability

The data presented in this study are available in the Appendix A.

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
