# Peer review of "Facultative Annual Life Cycles in Seagrasses"

_plants, 2023, doi:10.3390/plants12102002_

Round 1

Reviewer 1 Report

well structured manuscript, interesting research and topic, just some language imperfections attached

Author Response

Dear reviewer,

Thanks for your positive words. We changed all language improvements that you suggested, many thanks for providing them

Reviewer 2 Report

Manuscript ID: plants-2370476 Katwijk Title: Facultative annual life cycles in seagrasses

Authors: Marieke M. Van

Dear Authors

Article needs to be improved

You need to expand your resume. Indicate which specific annual plants are considered in the article. What is their areal distribution. Explain in more detail the system-wide “experiment.

As the authors write, there are only 6 annual marine plants and 63 perennials. Do we need to turn them into perennials?

The article lists 5 goals.

It is necessary in the Conclusion to write answers to these 5 goals.

It is necessary to add new literature for the last 5 years

Sincerely

21/04/2023

Minor editing of English language required

Author Response

Dear Reviewer,

Reviewer 3 Report

The manuscript provides a significant contribution to our knowledge about the behaviour of seagrasses. The main aim was to explore the relationships between environmental parameters and life cycle strategies.

The review suggests, that the annual vs perennial life cycles in facultative annual Z. marina and the other facultative annual seagrass species may be reversible.

The aims, sampling procedure, and methods of data analysis are clearly stated and introduced. The statistical analyses are appropriate. The results and facts are presented clearly and sufficiently fully and are separated from interpretations. The authors know well the literature on the subject and fairly discuss the correspondence of results.

Author Response

Dear Reviewer,

Many thanks for you positive evaluation.